# Development of selection strategies for genetic improvement in production traits of Mecheri sheep based on a Bayesian multi trait evaluation

Balakrishnan Balasundaram[1], Jaganadhan Muralidharan[2], Nagarajan Murali[3], Doraiswamy Cauveri[4], Angamuthu Raja[3], Moses Okpeku[5], Aranganoor Kannan Thiruvenkadan[3]*

1 Department of Animal Genetics and Breeding, Veterinary College and Research Institute, Tamil Nadu Veterinary and Animal Sciences University, Orathanadu, Tamil Nadu, India, 2 Mecheri Sheep Research Station, Tamil Nadu Veterinary and Animal Sciences University, Salem, Tamil Nadu, India, 3 Department of Animal Genetics and Breeding, Veterinary College and Research Institute, Tamil Nadu Veterinary and Animal Sciences University, Namakkal, Tamil Nadu, India, 4 Department of Animal Genetics and Breeding, Madras Veterinary College, Tamil Nadu Veterinary and Animal Sciences University, Chennai, Tamil Nadu, India, 5 Discipline of Genetics, School of Life Sciences, University of KwaZulu-Natal, Westville Campus, Durban, South Africa

* drthirusiva@gmail.com

**Data Availability Statement:** Data used in the analysis is anonymized and submitted as a

## Abstract

The progression of genetic selection techniques to enhance farm animal performance traits is guided by the present level of genetic variation and maternal impact in each trait, as well as the genetic association between traits. This study was conducted on a population of Mecheri sheep maintained from 1980 to 2018 at Mecheri Sheep Research Station, Pottaneri, India, to determine variance and covariance components, as well as genetic parameters for various production performance traits. A total of 2616 lambs, produced by 1044 dams and 226 sires, were examined in the study and the production traits of Mecheri sheep assessed include birth weight (BW), weaning weight (WW), six-month weight (SMW), nine-month weight (NMW), and yearling weight (YW). The Bayesian approach, using the Gibbs sampler, analyzed six animal models with different combinations of additive direct and maternal additive effects. Direct genetics, maternal genetics, and residual effects models were the major contributors to total phenotypic variation for all the production traits studied. Direct heritability estimates of birth weight, WW, SMW, NMW, and YW were 0.25, 0.20, 0.12, 0.14, and 0.13, respectively. The maternal heritability estimated for BW, WW, SMW, NMW, and YW were 0.17, 0.10, 0.12, 0.14, and 0.14, respectively. The maternal effects had a major impact on the pre-weaning production traits. The genetic correlations estimated between different pairs of production traits studied ranged from 0.19 to 0.93. The body weight at birth exhibited a higher genetic relationship with weaning weight than post-weaning growth characteristics, and the genetic correlation between weaning weight and post-weaning attributes was moderate to high (0.52 to 0.72). Based on the additive genetic variance in weaning weight and the correlation estimates of weaning weight with post-weaning

Supporting Information file along with the manuscript (S1 File).

**Funding:** The authors received no specific funding for this work.

**Competing interests:** The authors declare that they do not have any known competing financial interests or personal relationships that could appear to have influenced the work reported in this paper.

traits, weaning weight was proposed as a selection criterion for improving growth traits in Mecheri sheep.

## Introduction

Sheep farming, particularly with native breeds, plays vital roles in improving the livelihood and socio-economic status of a significant population of farmers in tropical countries [1]. Sheep farming in India is a source of consistent income for small, marginal, and landless farmers, allowing them to contribute to the country's economy. India has 44 distinct sheep breeds, making it a nation endowed with rich sheep population genetic diversity. Mecheri sheep is one of India's hairy sheep breeds, primarily raised for meat and skin production. The breed is renowned for its superior skin quality, higher dressing percentage, and reasonable growth rate in low-input environments. They are phenotypically polled, medium-sized, compact animals, and their natural breeding habitat is in the northwestern agro-climatic zone of Tamil Nadu. They are hardy animals that thrive well in semi-arid environments [2].

Mutton is the most commonly consumed red meat in India, and as such, it is highly demanded. However, the limited supply reduced the profits of sheep enterprises, and the slow growth rate of lambs is mainly responsible for the low profitability of sheep enterprises [3]. In general, the inefficiency of lambs for the performance of interrelated growth traits reflects the animals' adaptability, and it causes a major drain on the economy of sheep farming.

Genetic variance is the primary requirement for animal improvement through direct genetic selection. Along with direct additive genetic effects, maternal factors also play a role in shaping growth traits [4]. In many sheep breeds, both direct and maternal effects have been shown to have impacts on production traits [5–9]. However, the maternal effects are important, though often overlooked in genetic evaluation studies of tropical breeds, especially in small populations, such as Mecheri sheep, due to data limitations. If the maternal effect is disregarded, estimates of additive variance could skew upwards, and selection efficiency could be reduced [10]. Therefore, due consideration must be given to maternal genetic influences in breed improvement programs through genetic selection. In addition to direct selection, the identification of genetic correlations between economically significant growth factors may enable simultaneous improvement of associated traits. Frequency-based methods like restricted maximum likelihood (REML) are often used with a large number of observations to estimate high-precision covariance components and genetic correlations for performance traits in different sheep breeds [5, 11, 12]. However, Bayesian-based Gibbs sampling is a (co) variance component estimation approach that has several practical advantages over the REML method. The Bayesian approach does not specify regularity conditions on the probability model and always returns non-negative estimates for variance and the highest posterior density (HPD) interval [13]. The posterior probability distributions in Bayesian inference address the issue of uncertainty in unknown parameters and provide more precise estimates [6, 9]. This shows that for small populations, the Bayesian strategy is preferable to frequentist ones. In general, the population size, model, and technique used in calculating variance and covariance components all have an impact on how accurately genetic parameter estimations are made. The merit of this is that breeders are equipped with appropriate selection techniques for enhancing animal performance and with more accurate estimates of genetic parameters. In order to create the optimum genetic selection techniques for long-term improvement in these animals, the current work sets out to derive Bayesian estimates of (co)variance components

and genetic parameters for body weight variables in Mecheri lambs. The significance of additive genetic effects, genetic correlation, and maternal effects on lamb growth and population size were taken into account when doing this.

## Materials and methods

### Data

Data used in this study were obtained from the flock of Mecheri sheep kept at the Mecheri Sheep Research Station (MSRS) in Tamil Nadu, India, between 1980 and 2018. The farm is located at 11˚ 45' North latitude and 77˚ 56' East longitude, 650 feet above mean sea level. The information included the lambs' pedigree details as well as their birth and growth weights. Five economically important body weight traits of Mecheri sheep were studied at birth (BW), weaning or three months (WW), six months (SMW), nine months (NMW), and twelve months (YW). Table 1 summarizes the descriptive details of the dataset used in the investigation. All lambs in the study had identified sires and dams. A total of 2616 lambs, produced by 1044 dams and 226 sires, were examined in the study.

### Animal management

The Mecheri flock was semi-intensively managed, and the animals were housed in groups based on gender and age. Lambs were allowed to suckle for up to three months and were fed *ad libitum* concentrates from the second week of life until weaning, and the lambs were weaned at three months of age. Lambs were fed green fodder and 250 g of concentrate per day after weaning. Lambs were introduced to grazing for 7 to 8 hours per day when they were six months old. Animals were fed fodder and tree lopping in addition to grazing during the dry seasons, such as the summer. Controlled or hand mating has been performed to assure the recording of reliable pedigree data. The dams and the lambs were weighed at lambing, and each lamb's birth date, sex, and birth type were noted. At every growth stage, including weaning, six months, nine months, and a yearling, lambs were precisely weighed. Depending on their age, the majority of the ewes were slaughtered between the ages of six and seven.

### Statistical methods

The growth performance data were classified for the associated fixed effect, namely: sex of the lamb (two levels), season of birth (two levels: main and off), period of birth (eight levels: 1980–1984, 1985–1989, 1990–1994, 1995–1999, 2000–2004, 2005–2008, 2009–2013, and 2014–2018), type of birth (two levels: single and twin), and parity of the ewe (four levels: parity—1, 2, 3, and parity—4 and above). Initially, the general linear model procedure was used in the least squares analysis to determine the significant fixed effects for each performance trait. The analysis included the dam's weight at lambing as a covariate. The significant fixed factors identified in least-squares analyses were included in animal models for genetic analysis. Genetic analysis was carried out by fitting multivariate animal models [14] to estimate (co)variance

**Table 1. Data structure and description of body weight traits of Mecheri sheep.**

| Trait | Birth weight | Weaning weight | 6 month weight | 9 month weight | Yearling weight |
|---|---|---|---|---|---|
| Number of lambs have records | 2616 | 2286 | 1578 | 1203 | 1019 |
| Number of sires with records of progeny | 226 | 208 | 183 | 167 | 160 |
| Number of dams with records of progeny | 1044 | 961 | 814 | 701 | 638 |
| **Descriptive Statistics** | | | | | |
| Mean (kg) | 2.35 | 9.76 | 13.72 | 16.68 | 19.46 |
| Standard deviation | 0.46 | 2.34 | 2.64 | 2.86 | 3.08 |
| Coefficient of variation (%) | 19.18 | 24.7 | 19.77 | 17.58 | 16.15 |
| **Effect of fixed factors** | | | | | |
| Sex of the lamb | NS | NS | NS | * | ** |
| Birth type | ** | NS | NS | NS | NS |
| Parity of dam | * | NS | * | NS | NS |
| Period of birth | ** | ** | ** | ** | ** |
| Season of birth | NS | ** | NS | NS | * |
| Weight of ewes at lambing (covariate) | ** | ** | ** | ** | ** |

* P<0.01, Statistical significance of the source of variation in body weight traits

** P<0.05, Statistical significance of the source of variation in body weight traits

NS Not significant

components and corresponding genetic parameters of growth traits.

$$\textit{Model 1:} \quad y = Xb + Z_1a + e$$

$$\textit{Model 2:} \quad y = Xb + Z_1a + Z_2m + e \quad Cov(a, m) = 0$$

$$\textit{Model 3:} \quad y = Xb + Z_1a + Z_2m + e \quad Cov(a, m) = A\sigma_{am}$$

$$\textit{Model 4:} \quad y = Xb + Z_1a + Z_2pe + e$$

$$\textit{Model 5:} \quad y = Xb + Z_1a + Z_2m + Z_3pe + e \quad Cov(a, m) = 0$$

$$\textit{Model 6:} \quad y = Xb + Z_1a + Z_2m + Z_3pe + e \quad Cov(a, m) = A\sigma_{am}$$

Where $y$ is the vectors of records (N x 1), the vectors $b, a, m, pe$ and $e$ in the animal models indicate the fixed effects, additive genetic effects, maternal additive genetic effects, maternal permanent environmental effects and residual effects, respectively, and those vector's corresponding incidence matrices are $X, Z_1, Z_2,$ and $Z_3,$ respectively. The $A$ and $\sigma_{am}$ in the model indicated the numerator relationship matrix between animals and the covariance between direct and maternal genetic effects. The variance ($V$) and covariance ($Cov$) matrices, involving direct and maternal effects, were assumed as follows:

$$V(a) = A\sigma_a^2; \ V(m) = A\sigma_m^2; \ V(c) = I_d\sigma_{pe}^2; \ V(e) = I_n\sigma_e^2; \ and \ Cov(a, m) = A\sigma_{am}$$

$I_d$ and $I_n$ are identity matrices with orders equal to the number of dams and individual records, respectively. All of these systematic effects and (co)variance parts are assumed to have a uniform priori Gaussian distribution. The direct additive, maternal, permanent environmental, and residual variances are thought to have an inverse Wishart distribution [15].

To estimate (co)variance components, the Gibbs sampling method of Bayesian inference wasutilized, and the sample trace graphs were visually examined to verify convergence. This analysis was carried out in the GIBBS2F90 software program [16], and the variance components, namely: direct genetic $\left(\sigma_a^2\right)$, maternal genetic $\left(\sigma_m^2\right)$, covariance between direct and maternal genetic ($\sigma_{am}$), maternal permanent environmental $\left(\sigma_{pe}^2\right)$ and residual variances $\left(\sigma_e^2\right)$ were computed. The phenotypic covariance for each trait was computed as the sum of all variance components. Direct and maternal genetic correlation ($r_{am}$) was calculated as the ratio of the estimates of direct additive and maternal genetic covariance ($\sigma_{am}$) to the square root value of the product of $\sigma_a^2$ and $\sigma_m^2$. Direct heritability ($h^2$), maternal heritability $\left(h_m^2\right)$, and variance ratio estimate for maternal permanent environmental ($c^2$) were calculated as the ratio of the variance of random effects to the total phenotypic variance $\left(h^2 = \sigma_a^2 \,/\, \sigma_p^2;\ h_m^2 = \sigma_m^2 \,/\, \sigma_p^2;\ c^2 = \sigma_{pe}^2 \,/\, \sigma_p^2\right)$, as described by Misztal et al. [16]. In addition, tests using the log-likelihood ratio were carried out to identify which animal model best matched each trait [9].

## Results

### Descriptive statistics and fixed effects

The number of lambs studied and the arithmetic means for all the traits, along with pedigree details, are given in Table 1. The least-square means with standard deviation for the growth traits studied, namely, BW, WW, SMW, NMW, and YW, were 2.35 ± 0.46, 9.76 ± 2.34, 13.72 ± 2.64, 16.68 ± 2.86, and 19.46 ± 3.08 kg, respectively. The non-genetic factor, period of birth, and the covariate, dam body weight at lambing, significantly affected all the body weight traits studied. The effect of the sex of the lambs on NMW and YW; birth type on birth weight; parity order on BW and SMW; and season of birth on WW and YW were also found to be significant. Significant non-genetic factors were identified and included them as fixed factors in the genetic analysis.

### (Co)variance components and heritability estimates

The posterior marginal distributions of variance and covariance components followed the Gaussian distribution, and the symmetry in the distribution of the measures of central tendency supported the accuracy of the study. Because the samples utilized in the analysis to establish the genetic correlations were not distributed randomly, it is possible that the oscillations were stable. It was found that the study used a better burn-in duration, which facilitated convergence.

Tables 2–4 show the marginal posterior distributions of (co)variance components that were found using different models for body weight traits. It was found that animal model-2, which included direct additive and maternal genetic effects as random effects with zero covariance between them, was the most comprehensive model for all the traits studied. The additive genetic variance estimates for BW, WW, SMW, NMW, and YW were 0.055 ± 0.000, 0.983 ± 0.004, 0.933 ± 0.007, 1.872 ± 0.014, and 1.712 ± 0.014, respectively, and the corresponding maternal genetic effects for traits studied were 0.051 ± 0.0001, 0.419 ± 0.001, 0.839 ± 0.003, 1.643 ± 0.007, and 1.693 ± 0.007, respectively (Fig 1). The residual variances in body weight traits were comparatively low in the pre-weaning period (0.108–3.334), and they were higher (5.792–9.695) and increasing with age during the post-weaning period.

Table 5 lists the estimations of body weight characteristics such as maternal and direct heritabilities. All of the predicted direct heritabilities for body weight traits were within the

**Table 2. Marginal posterior distributions of (co)variance components of pre weaning body weight traits.**

| Trait | Model | (Co)variance component | Mean | Mode | Median | SD | CI 2.5% | CI 97.5% | Naive SE | Time Series SE |
|---|---|---|---|---|---|---|---|---|---|---|
| BW | Model-1 | $\sigma_a^2$ | 0.076 | 0.057 | 0.058 | 0.006 | 0.037 | 0.062 | 0.0001 | 0.0002 |
| | | $\sigma_e^2$ | 0.136 | 0.088 | 0.090 | 0.004 | 0.072 | 0.090 | 0.0001 | 0.0001 |
| | Model 2* | $\sigma_a^2$ | 0.055 | 0.038 | 0.037 | 0.005 | 0.018 | 0.040 | 0.0001 | 0.0004 |
| | | $\sigma_m^2$ | 0.051 | 0.033 | 0.033 | 0.003 | 0.017 | 0.031 | 0.0001 | 0.0002 |
| | | $\sigma_e^2$ | 0.108 | 0.091 | 0.090 | 0.004 | 0.073 | 0.089 | 0.0001 | 0.0002 |
| | Model 3 | $\sigma_a^2$ | 0.072 | 0.051 | 0.053 | 0.009 | 0.029 | 0.065 | 0.0002 | 0.0008 |
| | | $\sigma_m^2$ | 0.071 | 0.052 | 0.053 | 0.007 | 0.030 | 0.057 | 0.0001 | 0.0005 |
| | | $\sigma_{am}$ | 0.000 | -0.01 | -0.01 | 0.007 | -0.04 | -0.01 | 0.0002 | 0.0007 |
| | | $\sigma_e^2$ | 0.100 | 0.082 | 0.082 | 0.005 | 0.061 | 0.082 | 0.0001 | 0.0004 |
| | Model 4 | $\sigma_a^2$ | 0.045 | 0.036 | 0.036 | 0.008 | 0.021 | 0.043 | 0.0034 | 0.0036 |
| | | $\sigma_{pe}^2$ | 0.026 | 0.016 | 0.017 | 0.006 | 0.007 | 0.019 | 0.0034 | 0.0036 |
| | | $\sigma_e^2$ | 0.096 | 0.086 | 0.087 | 0.007 | 0.074 | 0.090 | 0.0034 | 0.0035 |
| | Model 5 | $\sigma_a^2$ | 0.056 | 0.038 | 0.038 | 0.005 | 0.018 | 0.040 | 0.0001 | 0.0003 |
| | | $\sigma_m^2$ | 0.042 | 0.023 | 0.024 | 0.003 | 0.008 | 0.022 | 0.0001 | 0.0004 |
| | | $\sigma_{pe}^2$ | 0.037 | 0.01 | 0.019 | 0.003 | 0.004 | 0.016 | 0.0001 | 0.0003 |
| | | $\sigma_e^2$ | 0.107 | 0.088 | 0.089 | 0.004 | 0.071 | 0.087 | 0.0001 | 0.0002 |
| | Model 6 | $\sigma_a^2$ | 0.067 | 0.046 | 0.049 | 0.007 | 0.027 | 0.056 | 0.0001 | 0.0006 |
| | | $\sigma_m^2$ | 0.056 | 0.035 | 0.037 | 0.007 | 0.014 | 0.043 | 0.0001 | 0.0007 |
| | | $\sigma_{am}$ | 0.005 | -0.01 | -0.01 | 0.006 | -0.03 | -0.01 | 0.0001 | 0.0006 |
| | | $\sigma_{pe}^2$ | 0.039 | 0.022 | 0.021 | 0.003 | 0.005 | 0.018 | 0.0001 | 0.0004 |
| | | $\sigma_e^2$ | 0.100 | 0.082 | 0.082 | 0.004 | 0.064 | 0.082 | 0.0001 | 0.0003 |
| WW | Model 1 | $\sigma_a^2$ | 1.096 | 1.031 | 1.036 | 0.208 | 0.684 | 1.470 | 0.0024 | 0.0082 |
| | | $\sigma_e^2$ | 3.504 | 3.439 | 3.460 | 0.172 | 3.103 | 3.782 | 0.0020 | 0.0059 |
| | Model 2* | $\sigma_a^2$ | 0.983 | 0.946 | 0.949 | 0.225 | 0.564 | 1.424 | 0.0045 | 0.0189 |
| | | $\sigma_m^2$ | 0.419 | 0.415 | 0.398 | 0.095 | 0.210 | 0.594 | 0.0019 | 0.0073 |
| | | $\sigma_e^2$ | 3.334 | 3.278 | 3.316 | 0.171 | 2.976 | 3.632 | 0.0034 | 0.0108 |
| | Model 3 | $\sigma_a^2$ | 1.980 | 1.783 | 1.903 | 0.484 | 1.155 | 3.066 | 0.0097 | 0.0522 |
| | | $\sigma_m^2$ | 1.089 | 1.080 | 1.060 | 0.216 | 0.678 | 1.510 | 0.0043 | 0.0182 |
| | | $\sigma_{am}$ | -1.10 | -1.03 | -1.09 | 0.290 | -1.79 | -0.65 | 0.0058 | 0.0322 |
| | | $\sigma_e^2$ | 2.881 | 2.891 | 2.880 | 0.268 | 2.282 | 3.328 | 0.0054 | 0.0266 |
| | Model 4 | $\sigma_a^2$ | 0.908 | 0.886 | 0.886 | 0.217 | 0.526 | 1.366 | 0.0076 | 0.0196 |
| | | $\sigma_{pe}^2$ | 0.208 | 0.198 | 0.195 | 0.074 | 0.075 | 0.352 | 0.0047 | 0.0099 |
| | | $\sigma_e^2$ | 3.302 | 3.262 | 3.293 | 0.173 | 2.943 | 3.616 | 0.0067 | 0.0131 |
| | Model 5 | $\sigma_a^2$ | 0.918 | 0.887 | 0.887 | 0.214 | 0.523 | 1.363 | 0.0043 | 0.0163 |
| | | $\sigma_m^2$ | 0.326 | 0.289 | 0.301 | 0.086 | 0.151 | 0.490 | 0.0017 | 0.0104 |
| | | $\sigma_{pe}^2$ | 0.219 | 0.200 | 0.197 | 0.071 | 0.072 | 0.349 | 0.0014 | 0.0066 |
| | | $\sigma_e^2$ | 3.313 | 3.264 | 3.295 | 0.169 | 2.940 | 3.613 | 0.0034 | 0.0098 |
| | Model 6 | $\sigma_a^2$ | 2.097 | 1.870 | 2.02 | 0.544 | 1.135 | 3.275 | 0.0109 | 0.066 |
| | | $\sigma_m^2$ | 0.988 | 0.945 | 0.961 | 0.251 | 0.513 | 1.465 | 0.0050 | 0.0231 |
| | | $\sigma_{am}$ | -1.15 | -1.08 | -1.15 | 0.342 | -1.91 | -0.56 | 0.0069 | 0.0421 |
| | | $\sigma_{pe}^2$ | 0.276 | 0.251 | 0.254 | 0.079 | 0.105 | 0.415 | 0.0016 | 0.0071 |
| | | $\sigma_e^2$ | 2.731 | 2.717 | 2.726 | 0.299 | 2.063 | 3.242 | 0.0060 | 0.0310 |

BW, birth weight; WW, weaning weight; CI, credibility interval; SD, standard deviation; SE, standard error.

* Best fit model.

**Table 3. Marginal posterior distributions of (co)variance components of 6 and 9-months body weight traits.**

| Trait | Model | (Co)variance component | Mean | Mode | Median | SD | CI 2.5% | CI 97.5% | Naive SE | Time Series SE |
|---|---|---|---|---|---|---|---|---|---|---|
| SMW | Model 1 | $\sigma^2_a$ | 1.550 | 1.457 | 1.483 | 0.390 | 0.763 | 2.329 | 0.004 | 0.018 |
| | | $\sigma^2_e$ | 5.824 | 5.783 | 5.776 | 0.341 | 5.082 | 6.437 | 0.003 | 0.013 |
| | Model 2* | $\sigma^2_a$ | 0.933 | 0.785 | 0.872 | 0.377 | 0.306 | 1.741 | 0.007 | 0.051 |
| | | $\sigma^2_m$ | 0.839 | 0.820 | 0.811 | 0.185 | 0.487 | 1.218 | 0.003 | 0.013 |
| | | $\sigma^2_e$ | 5.792 | 5.736 | 5.772 | 0.319 | 5.149 | 6.385 | 0.006 | 0.026 |
| | Model 3 | $\sigma^2_a$ | 2.219 | 1.652 | 2.156 | 0.827 | 0.763 | 4.010 | 0.016 | 0.106 |
| | | $\sigma^2_m$ | 1.302 | 1.130 | 1.24 | 0.362 | 0.690 | 2.105 | 0.007 | 0.035 |
| | | $\sigma_{am}$ | -1.05 | -0.956 | -1.024 | 0.500 | -2.235 | -0.26 | 0.010 | 0.055 |
| | | $\sigma^2_e$ | 5.217 | 5.232 | 5.223 | 0.483 | 4.201 | 6.051 | 0.009 | 0.054 |
| | Model 4 | $\sigma^2_a$ | 0.952 | 0.775 | 0.894 | 0.346 | 0.379 | 1.748 | 0.010 | 0.043 |
| | | $\sigma^2_{pe}$ | 0.299 | 0.281 | 0.282 | 0.110 | 0.112 | 0.524 | 0.005 | 0.013 |
| | | $\sigma^2_e$ | 5.701 | 5.711 | 5.701 | 0.302 | 5.064 | 6.262 | 0.009 | 0.023 |
| | Model 5 | $\sigma^2_a$ | 0.962 | 0.777 | 0.896 | 0.342 | 0.376 | 1.745 | 0.006 | 0.040 |
| | | $\sigma^2_m$ | 0.715 | 0.668 | 0.687 | 0.171 | 0.371 | 1.047 | 0.003 | 0.013 |
| | | $\sigma^2_{pe}$ | 0.310 | 0.283 | 0.284 | 0.107 | 0.109 | 0.521 | 0.002 | 0.010 |
| | | $\sigma^2_e$ | 5.711 | 5.712 | 5.703 | 0.298 | 5.061 | 6.259 | 0.006 | 0.020 |
| | Model 6 | $\sigma^2_a$ | 1.559 | 1.308 | 1.431 | 0.592 | 0.632 | 2.883 | 0.011 | 0.076 |
| | | $\sigma^2_m$ | 0.999 | 0.950 | 0.950 | 0.289 | 0.499 | 1.617 | 0.005 | 0.033 |
| | | $\sigma_{am}$ | -0.70 | -0.59 | -0.65 | 0.356 | -1.65 | -0.22 | 0.007 | 0.047 |
| | | $\sigma^2_{pe}$ | 0.369 | 0.319 | 0.338 | 0.129 | 0.127 | 0.642 | 0.002 | 0.012 |
| | | $\sigma^2_e$ | 5.474 | 5.454 | 5.475 | 0.381 | 4.633 | 6.134 | 0.007 | 0.037 |
| NMW | Model 1 | $\sigma^2_a$ | 2.047 | 1.894 | 1.952 | 0.628 | 0.894 | 3.310 | 0.007 | 0.039 |
| | | $\sigma^2_e$ | 10.19 | 10.24 | 10.16 | 0.607 | 8.961 | 11.34 | 0.007 | 0.026 |
| | Model 2* | $\sigma^2_a$ | 1.872 | 1.349 | 1.777 | 0.744 | 0.751 | 3.450 | 0.014 | 0.084 |
| | | $\sigma^2_m$ | 1.643 | 1.532 | 1.582 | 0.361 | 1.028 | 2.445 | 0.007 | 0.024 |
| | | $\sigma^2_e$ | 9.434 | 9.485 | 9.448 | 0.604 | 8.220 | 10.52 | 0.012 | 0.049 |
| | Model 3 | $\sigma^2_a$ | 4.524 | 4.314 | 4.467 | 1.544 | 1.810 | 7.706 | 0.030 | 0.179 |
| | | $\sigma^2_m$ | 2.810 | 2.581 | 2.722 | 0.727 | 1.511 | 4.365 | 0.014 | 0.071 |
| | | $\sigma_{am}$ | -2.51 | -2.43 | -2.490 | 0.959 | -4.61 | -0.82 | 0.019 | 0.107 |
| | | $\sigma^2_e$ | 8.259 | 8.205 | 8.241 | 0.889 | 6.492 | 9.913 | 0.017 | 0.096 |
| | Model 4 | $\sigma^2_a$ | 1.707 | 1.509 | 1.611 | 0.646 | 0.670 | 3.198 | 0.016 | 0.075 |
| | | $\sigma^2_{pe}$ | 0.501 | 0.391 | 0.470 | 0.203 | 0.171 | 0.9672 | 0.007 | 0.025 |
| | | $\sigma^2_e$ | 9.422 | 9.339 | 9.400 | 0.583 | 8.222 | 10.56 | 0.014 | 0.044 |
| | Model 5 | $\sigma^2_a$ | 1.718 | 1.511 | 1.613 | 0.643 | 0.667 | 3.195 | 0.012 | 0.072 |
| | | $\sigma^2_m$ | 1.484 | 1.396 | 1.439 | 0.365 | 0.818 | 2.237 | 0.007 | 0.029 |
| | | $\sigma^2_{pe}$ | 0.511 | 0.393 | 0.471 | 0.2 | 0.168 | 0.964 | 0.004 | 0.022 |
| | | $\sigma^2_e$ | 9.432 | 9.340 | 9.402 | 0.580 | 8.219 | 10.56 | 0.011 | 0.041 |
| | Model 6 | $\sigma^2_a$ | 4.411 | 3.768 | 4.161 | 1.742 | 1.751 | 8.453 | 0.034 | 0.218 |
| | | $\sigma^2_m$ | 2.276 | 2.117 | 2.176 | 0.653 | 1.175 | 3.706 | 0.013 | 0.072 |
| | | $\sigma_{am}$ | -2.21 | -1.93 | -2.10 | 0.946 | -4.38 | -0.73 | 0.018 | 0.120 |
| | | $\sigma^2_{pe}$ | 0.588 | 0.586 | 0.557 | 0.232 | 0.198 | 1.083 | 0.004 | 0.024 |
| | | $\sigma^2_e$ | 8.248 | 8.655 | 8.319 | 1.003 | 6.018 | 9.958 | 0.020 | 0.119 |

SMW, six month body weight; NMW, nine month body weight; CI, credibility interval; SD, standard deviation; SE, standard error.

* Best fit model.

**Table 4. Marginal posterior distributions of (Co)variance components of yearling weight (YW).**

| Model | (Co)variance component | Mean | Mode | Median | SD | CI 2.5% | CI 97.5% | Naive SE | Time Series SE |
|---|---|---|---|---|---|---|---|---|---|
| Model 1 | $\sigma_a^2$ | 1.949 | 1.794 | 1.858 | 0.651 | 0.783 | 3.249 | 0.008 | 0.038 |
| | $\sigma_e^2$ | 10.53 | 10.47 | 10.48 | 0.673 | 9.170 | 11.80 | 0.008 | 0.026 |
| Model 2* | $\sigma_a^2$ | 1.713 | 1.522 | 1.595 | 0.702 | 0.724 | 3.237 | 0.014 | 0.081 |
| | $\sigma_m^2$ | 1.694 | 1.596 | 1.640 | 0.372 | 1.032 | 2.465 | 0.007 | 0.024 |
| | $\sigma_e^2$ | 9.696 | 9.805 | 9.692 | 0.645 | 8.385 | 10.94 | 0.013 | 0.045 |
| Model 3 | $\sigma_a^2$ | 4.096 | 4.029 | 4.027 | 1.413 | 1.532 | 7.169 | 0.028 | 0.181 |
| | $\sigma_m^2$ | 2.809 | 2.469 | 2.696 | 0.815 | 1.430 | 4.591 | 0.016 | 0.074 |
| | $\sigma_{am}$ | -2.43 | -1.94 | -2.33 | 1.029 | -4.73 | -0.62 | 0.021 | 0.132 |
| | $\sigma_e^2$ | 8.790 | 8.577 | 8.766 | 0.857 | 7.018 | 10.40 | 0.017 | 0.086 |
| Model 4 | $\sigma_a^2$ | 1.651 | 1.248 | 1.563 | 0.695 | 0.604 | 3.188 | 0.017 | 0.088 |
| | $\sigma_{pe}^2$ | 0.671 | 0.561 | 0.625 | 0.261 | 0.264 | 1.250 | 0.008 | 0.029 |
| | $\sigma_e^2$ | 9.570 | 9.640 | 9.571 | 0.667 | 8.276 | 10.89 | 0.017 | 0.048 |
| Model 5 | $\sigma_a^2$ | 1.662 | 1.250 | 1.565 | 0.692 | 0.601 | 3.185 | 0.014 | 0.085 |
| | $\sigma_m^2$ | 1.406 | 1.272 | 1.359 | 0.354 | 0.759 | 2.188 | 0.007 | 0.027 |
| | $\sigma_{pe}^2$ | 0.681 | 0.563 | 0.626 | 0.257 | 0.261 | 1.247 | 0.005 | 0.026 |
| | $\sigma_e^2$ | 9.581 | 9.641 | 9.573 | 0.664 | 8.273 | 10.89 | 0.013 | 0.045 |
| Model 6 | $\sigma_a^2$ | 3.136 | 2.646 | 2.932 | 1.221 | 1.339 | 6.027 | 0.024 | 0.148 |
| | $\sigma_m^2$ | 1.720 | 1.572 | 1.650 | 0.490 | 0.881 | 2.820 | 0.010 | 0.046 |
| | $\sigma_{am}$ | -1.46 | -1.02 | -1.40 | 0.659 | -3.10 | -0.52 | 0.013 | 0.073 |
| | $\sigma_{pe}^2$ | 0.749 | 0.596 | 0.698 | 0.277 | 0.268 | 1.337 | 0.006 | 0.028 |
| | $\sigma_e^2$ | 9.055 | 9.056 | 9.076 | 0.814 | 7.278 | 10.506 | 0.016 | 0.077 |

CI, credibility interval; SD, standard deviation; SE, standard error.

* Best fit model.

moderate (0.12–0.25) range. For BW, WW, SMW, NMW, and YW, the estimates of maternal heritability were 0.17, 0.10, 0.12, 0.14, and 0.14, respectively.

## Genetic correlation

Table 6 and Fig 2 show the predicted direct genetic correlations among growth traits. All of the estimated genotypic correlations between body weight variables werefavorable. In contrast to other body weight variables, which ranged in correlation from 0.19 to 0.27, birth weight showed a weak genetic connection with weaning weight (0.65). Weaning weight had a strong genetic association with SMW (0.72) and a weak genetic correlation with NMW (0.60) and YW (0.52). Estimates of genetic correlations between SMW and subsequent body weight traits were found to be very high (0.93 and 0.81). A similar strong genetic relationship was encountered between NMW and the subsequent body weight trait (0.91).

## Discussion

### Coefficient of variance

In this study, the number of lambs from birth to various growth stages decreased as age increased. This is because lambscull, through the sale of ewes and rams for breeding purposes or the death of lambs. The coefficients of variation for body weight traits in Mecheri sheep ranged from 16.15 to 24.5%, with 3-month body weight having the highest coefficient. The high coefficient of variation observed in weaning weight indicated that the environmental influence

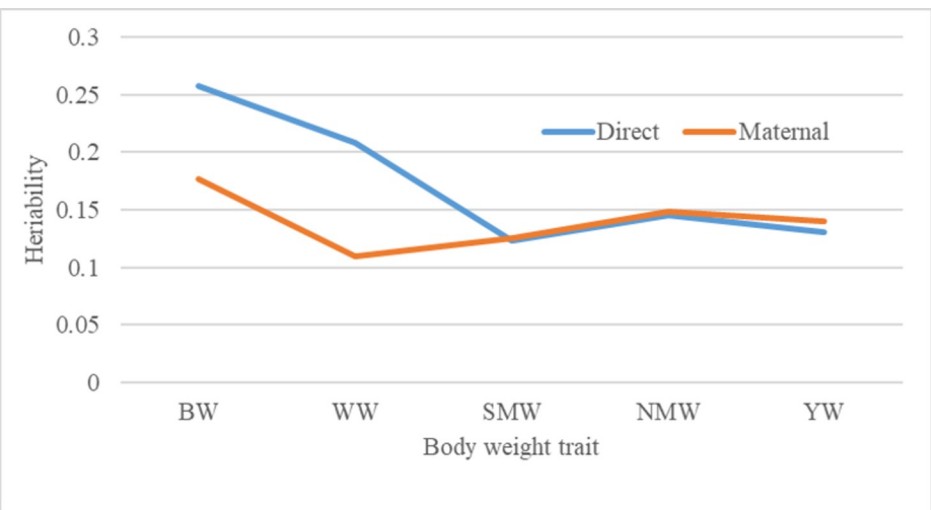

**Fig 1. Additive direct and maternal heritability estimates of growth traits in Mecheri sheep.** High additive genetic variance in pre-weaning growth traits. The level of maternal additive heritability present in birth weight indicated the importance of maternal genes in determining lamb birth weight.

on weaning was positive and incremental. The coefficients of variation observed were within reported ranges in the Lori, Nellore, and Iran Black breeds [8, 9, 17].

## Body weight trait covariance components and heritability estimates

**Birth weight.** Birth weight is usually measured and used as an indication of growth potential. The maternal genetic influence and direct additive in model 2 were both sufficient to account for birth weight variation. For body weight features in lambs of various sheep breeds, maternal impacts and direct genetic effects have been reported to be significant [18–20]. The direct heritability ($h^2$) estimate of birth weight from the most comprehensive model was 0.26. It was comparable with reported findings in various tropical sheep breeds [11, 21–23]. However, this estimate was comparatively higher than those estimated in Malapura [6], Nilagiri [7],

**Table 5. Marginal posterior distributions of additive direct and maternal heritabilities for growth traits.**

| Trait | Mean | Mode | Median | SD | CI 2.5% | CI 97.5% | Naïve SE | Time Series SE |
|---|---|---|---|---|---|---|---|---|
| **Direct heritability** | | | | | | | | |
| BW | 0.258 | 0.221 | 0.218 | 0.038 | 0.139 | 0.290 | 0.001 | 0.003 |
| WW | 0.208 | 0.209 | 0.213 | 0.044 | 0.127 | 0.297 | 0.001 | 0.004 |
| SMW | 0.123 | 0.120 | 0.127 | 0.048 | 0.044 | 0.226 | 0.001 | 0.006 |
| NMW | 0.145 | 0.150 | 0.149 | 0.054 | 0.062 | 0.261 | 0.001 | 0.006 |
| YW | 0.131 | 0.138 | 0.137 | 0.051 | 0.060 | 0.245 | 0.001 | 0.006 |
| **Maternal heritability** | | | | | | | | |
| BW | 0.177 | 0.184 | 0.186 | 0.024 | 0.131 | 0.226 | 0.001 | 0.001 |
| WW | 0.110 | 0.097 | 0.094 | 0.020 | 0.048 | 0.125 | 0.000 | 0.002 |
| SMW | 0.125 | 0.111 | 0.118 | 0.024 | 0.067 | 0.158 | 0.001 | 0.002 |
| NMW | 0.148 | 0.131 | 0.133 | 0.026 | 0.081 | 0.184 | 0.001 | 0.002 |
| YW | 0.140 | 0.133 | 0.136 | 0.027 | 0.081 | 0.184 | 0.001 | 0.002 |

BW, birth weight; WW, weaning weight; SMW, six month body weight; NMW, nine month body weight; YW, yearling weight; CI, credibility interval; SD, standard deviation; SE, standard error.

**Table 6. Marginal posterior distributions of genetic covariance and genetic correlations among body weight traits.**

| Trait(s) | Mean | Mode | Median | SD | CI 2.5% | CI 97.5% | Naïve SE | Time Series SE |
|---|---|---|---|---|---|---|---|---|
| **Genetic covariance** | | | | | | | | |
| BW-WW | 0.129 | 0.113 | 0.111 | 0.027 | 0.0496 | 0.156 | 0.000 | 0.002 |
| BW-SMW | 0.068 | 0.049 | 0.048 | 0.037 | -0.02 | 0.122 | 0.000 | 0.004 |
| BW-NMW | 0.083 | 0.053 | 0.061 | 0.050 | -0.04 | 0.160 | 0.001 | 0.005 |
| BW-YW | 0.066 | 0.054 | 0.047 | 0.050 | -0.05 | 0.145 | 0.001 | 0.005 |
| WW-SMW | 0.680 | 0.600 | 0.637 | 0.264 | 0.198 | 1.247 | 0.005 | 0.030 |
| WW-NMW | 0.785 | 0.638 | 0.739 | 0.345 | 0.163 | 1.506 | 0.006 | 0.037 |
| WW-YW | 0.668 | 0.521 | 0.612 | 0.344 | 0.051 | 1.386 | 0.006 | 0.038 |
| SMW-NMW | 1.198 | 1.169 | 1.142 | 0.486 | 0.417 | 2.242 | 0.009 | 0.059 |
| SMW-YW | 1.016 | 0.862 | 0.944 | 0.452 | 0.318 | 1.965 | 0.009 | 0.058 |
| NMW-YW | 1.601 | 1.562 | 1.509 | 0.695 | 0.571 | 3.094 | 0.013 | 0.080 |
| **Genetic correlation** | | | | | | | | |
| BW-WW | 0.654 | 0.670 | 0.648 | 0.112 | 0.377 | 0.817 | 0.002 | 0.008 |
| BW-SMW | 0.275 | 0.283 | 0.264 | 0.206 | -0.17 | 0.617 | 0.004 | 0.022 |
| BW-NMW | 0.264 | 0.280 | 0.255 | 0.208 | -0.22 | 0.603 | 0.004 | 0.022 |
| BW-YW | 0.193 | 0.242 | 0.192 | 0.226 | -0.29 | 0.569 | 0.005 | 0.025 |
| WW-SMW | 0.726 | 0.785 | 0.742 | 0.145 | 0.320 | 0.886 | 0.003 | 0.020 |
| WW-NMW | 0.595 | 0.644 | 0.595 | 0.172 | 0.161 | 0.835 | 0.003 | 0.020 |
| WW-YW | 0.523 | 0.572 | 0.526 | 0.194 | 0.054 | 0.810 | 0.004 | 0.022 |
| SMW-NMW | 0.932 | 0.943 | 0.924 | 0.052 | 0.767 | 0.974 | 0.001 | 0.005 |
| SMW-YW | 0.818 | 0.852 | 0.815 | 0.112 | 0.542 | 0.969 | 0.002 | 0.014 |
| NMW-YW | 0.910 | 0.931 | 0.912 | 0.075 | 0.692 | 0.975 | 0.002 | 0.011 |

BW, birth weight; WW, weaning weight; SMW, six month body weight; NMW, nine month body weight; YW, yearling weight; CI, credibility interval; SD, standard deviation; SE, standard error.

Corriedale [12], Kermani [24], Chokla [25], and Nellore [26] lambs. Higher heritability in Mecheri sheep was previously reported [27], in a study in which maternal effects evaluation was absent. In this study, the moderate heritability of birth weight was attributed to low environmental change as a result of improved feeding, health, and shelter management practices for pregnant animals. Higher body weight at birth was considered an early indicator for lambs with potential economic value. Furthermore, improving the trait may help minimize lamb mortality, as underweight lambs are a common cause of death in the first few days of life [28]. However, avoiding complications such as dystocia in dams requires optimizingthe birth weight. The performance of Mecheri lambs studied for birth weight, as well as the estimated additive genetic variance and direct heritability estimates, indicated that there is room for improvement in the trait.

Maternal additive genetic influences accounted for 17.7% of total phenotypic variability in birth weight, indicating the importance of maternal genes in determining lamb birth weight. Maternal genes influence the uterine environment, uterine capacity, and quality for fetal growth and, thus, lamb birth weight [29]. Previous findings in several sheep breeds in India [25, 26, 30, 31] were similar to the estimated maternal heritability. However, higher estimates were reported for Kermani [24] and Malpura [29] lambs. Models 3 and 6 produced positive but unremarkable estimates of covariance between additive direct and maternal effects on birth weight. The results showed that the selection of lambs for an additive direct genetic influence on birth weight had no effect on the maternal potential of those lambs.

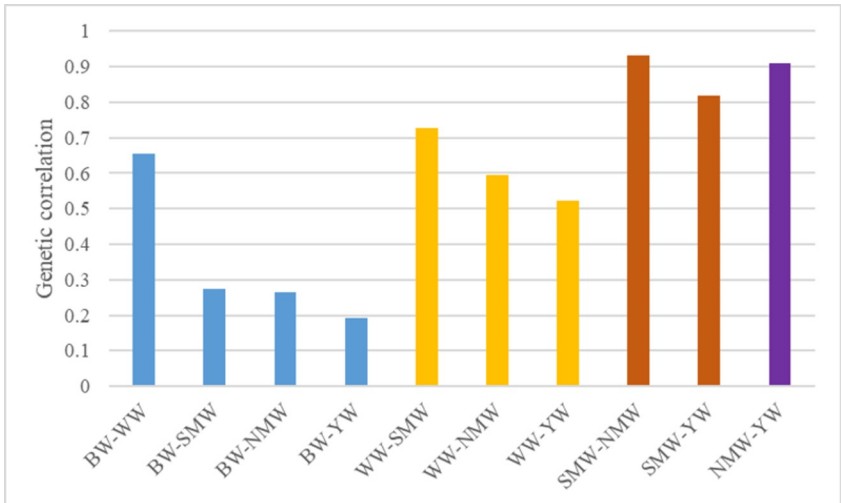

**Fig 2. Genetic correlation estimates between growth traits of Mecheri sheep.** A moderate to strong genetic association between weaning weight and other body weight traits and a high genetic relationship between succeeding traits.

**Weaning weight.** Similar to birth weight, maternal factors and direct additive effects both had a significant impact on the overall phenotypic variance in weaning weight. Similar results were seen in Chokla [25], Nellore [31], Targhee [32], Malpura [33], and other sheep breeds [34]. The estimate from the current study was less than that for Nellore [9] and Malpura [6, 22] sheep. Weaning weight in various sheep breeds was estimated to have lower heritability values [7, 26, 35]. Lamb weight at weaning had a lower heritability than lamb birth weight. This reduction confirmed that the environmental variation increased after birth. During the pre-weaning period, the fluctuations in environmental variation could be attributable to the nutritional level of ewes, the varying level of milk production of ewes, and the feeding or grazing behavior of developing lambs. The estimated moderate heritability in WW indicated that selecting animals based on body weight at weaning could result in better genetic gain.

In this study, the maternal heritability found in weaning weight was moderate (0.11), which is about the same as what was found in Kermani [24] but lower than what was found in Malpura [22, 29] and Nellore [26]. Pre-weaning body weight traits' predicted heritability showed that the maternal additive genetic contribution was greatest at birth and waned as we got closer to weaning. As lambsage, it is anticipated that the maternal genetic effects visible during pregnancy and lactation will lessen. The present findings support Robison's [36] conclusion that mammalian maternal impact is greatest in young animals but weakens with age. The significant reduction in the maternal genetic effect was observed throughout the pre-weaning growth stage, and this observation is consistent with the findings reported in Swedish Finewool [37], Afrino [18], Kermani [24], Chokla [25], and Baluchi [5, 38] sheep breeds. In contrast, during the pre-weaning phase, Lori and Nellore lambs showed a rising trend for maternal heritabilities, according to earlier reports [8, 9]. Maternal permanent environmental effects on preweaning traits in the present study were not substantial. On the other hand, it was reported that maternal permanent environmental effects significantly contributed to the phenotypic variability in the Indian Nellore and Iranian Baluchi sheep populations [5, 9].

**Post-weaning growth traits.** The absence of maternal effects or the addition of permanent environmental effects to model 2 had no effect on the total phenotypic variance in all post-weaning body weight traits. For SMW, NMW, and YW, the additive direct heritability

estimated from the best model (model 2) was 0.12, 0.14, and 0.13, respectively. The reported heritability estimates for 6-month body weight in the sheep breeds Chokla, Iran Black, Nilagiri, and Nellore are comparable to the current results [7, 17, 31]. Higher heritability estimates for SMW were reported in Lori [8], Nellore [9], Malapura [29], and Sangsari [39] sheep breeds, but low estimates were reported in Baluchi [5] and Sanjabi [40] lambs. According to past findings in several sheep breeds, the heritability estimates for body weight at nine months and 12 months in Mecheri sheep are accurate [7, 26, 29, 30]. The current estimates, however, were lower than those of different researchers in different sheep breeds [8, 9, 25]. Research on growth traits has established age-related increase in direct heritability for body weight [24, 41]. In contrast, the heritability estimates for post-weaning traits in this study were lower than those for pre-weaning traits. Similar results have been reported in the Iranian Sangsari [39] and Indian Nellore [9] sheep breeds. Variations in heritability estimates reported in different research studies may be due to breed variances, different analytical approaches, and influencing environmental factors. The present study estimated moderate heritability values for post-weaning growth traits. This demonstrates there is adequate genetic variability for improved post-weaning growth through genetic selection.

The reduction of genetic variance during the post-weaning period could be attributable to the generation of high environmental variance due to prolonged grazing during the post-weaning period. The varying nature of the prevailing climatic conditions in the study area and external factors influenced the quantity and quality of pasture on the grazing land, which eventually increased the environmental variance. Post-weaning lambs exposed to poor range circumstances may not express their full genetic potential, resulting in lower heritability [42]. Post-weaning characteristics with moderate heritability estimates indicate the presence of genetic diversity that may be tapped through genetic selection to enhance those traits.

In the current study, maternal genes had a significant influence on all post-weaning traits. According to the moderate maternal heritability for post-weaning body weight features, maternal genes play a significant role in lamb phenotypic expression. In general, maternal effects on lamb body weight traits diminished as the animals grew older [43–45]. In this study, NMW and YW maternal heritability estimates were higher than SMW, and similar findings were reported in different sheep breeds [7, 38, 46]. The maternal genetic effects began during pregnancy and became more pronounced during the weaning stage. Later, the post-weaning lambs carried over the long-lasting maternal effects from the prenatal and preweaning stages, combining them with the powerful additive and environmental effects to express their phenotype. The models that included permanent environmental effects were not able to fully explain the differences in phenotypes in all post-weaning traits. This suggests that permanent environmental effects are insignificant.

In the model that included the direct-maternal genetic relationship, the estimated covariance between additive direct and maternal additive genetic components was negative for all postweaning traits, and similar findings have been reported in Malapura, Nellore, and Corriedale sheep [6, 12]. If animals are selected on the basis of direct additive genetic effects, the antagonistic correlation between additive and maternal additive components may degrade their maternal contribution. This antagonistic relationship between additive and maternal additive effects of important fixed factors [47], the degree of dependency of parameters on the number of offspring produced per dam [48], and the presence of greater variance between sire and dams [49] attest to this. However, the reported unfavorable correlation between additive and maternal additive effects is frequently due to statistical processes rather than a biological cause [50]. It is also possible that nature uses the competition between direct additive and maternal additive genes for body weight traits to optimize phenotypes at intermediate values [43]. This is regarded as one of the methods for preserving genetic diversity [51]. A lot of

different sheep breeds have shown the importance of the interaction between direct additive genes and maternal additive genes to determine growth traits [6, 8, 9]. However, when a direct additive-maternal additive correlation was added to the additive and maternal effects, the current research demonstrated no significance.

### Genetic correlation between growth traits

The genetic correlation estimates computed from the multivariate analysis among different production traits ranged from 0.19 to 0.93. There was a strong positive genetic correlation (0.65) between birth weight and weaning weight, indicating that selecting for birth weight could result in an improvement in weaning weight. The estimated genetic correlation between BW and WW in this study was higher than that reported in the Red Maasai (0.54), Columbia (0.56), and Corridale (0.43) sheep populations [23, 52, 53]. Birth weight and other body weight traits had lower genetic correlations than weaning weight and other body weight traits. It has been suggested that using birth weight as an indirect selection criterion for genetic improvement in post-weaning body weight characteristics may not yield the desired results. The genotypic correlations between birth weights as well as other body weight traits demonstrated a pattern in which estimates decreased as the difference in age lengthened.

Weaning weight had a strong genetic correlation with body weight at six months and moderate correlations with body weight at nine and twelve months in Mecheri sheep. The genetic correlation observed between weaning weight and six-month weight in the current study was consistent with the findings in Iranian Baluchi breed [5] and Kordi [54] sheep. Nellore [9] and Corriedale [53] sheep had higher estimates. Lower estimates between the same traits in Doyogena [1] and Lori [8] lambs were also reported. The moderate to strong genetic correlations observed in this study for weaning weight suggested that selecting for the trait could result in a better response to later body weight traits. Weaning lamb selection saves time and resources while progressing towards genetic improvement in market weight.

The genetic correlations encountered for body weight at six and nine months with weight at later ages were positive and very strong, indicating that if the animals were selected at six or nine months of age, there would be a very good response in body weight at later ages. Other body weight traits, like birth weight, showed a decreasing trend in their genetic relationship with age advancement. Previously, similar findings in various sheep breeds were reported [9, 23, 40, 53].

The present genetic relationship assessment between body weight traits did not show any antagonistic associations. Similar results were reported earlier in different sheep populations [8, 24, 31]. It was hypothesized that choosing any of the early body weights studied would result in positive responses to later-age traits. However, taking into account the level of genetic variance in different body weight traits as well as their genetic correlation estimates, weaning weight was proposed as a selection criterion for improving later-age growth traits in the studied population. The genetic parameters estimated using the Bayesian Gibbs sampler in this study were similar to or very close to previously reported REML-based results in the same breed [55]. It proved that the estimates made with REML for genetic parameters were accurate and confirmed the size and distribution of the Mecheri sheep population that had been studied before.

## Conclusions

The current study found that an animal model with additive direct effects and maternal additive effects is adequate for explaining variation in Mecheri sheep body weight traits. The genetic variance was greatest in preweaning growth traits in the Mecheri sheep population studied and lowest in post-weaning traits. The decrease in direct heritability in the post-

weaning stage could be attributed to increased environmental variance after weaning. The heritability estimates of growth traits confirmed the availability of genetic variation in body weight traits for animal genetic improvement. Furthermore, the moderate heritability estimates in weaning weight suggested that selecting animals based on three-month body weight could result in better genetic gain. The maternal additive genetic variability for birth weight in Mecheri sheep was moderate, and the values suggested that maternal genes are important in determining lamb birth weight. Genetic correlations between successive body weight traits are always stronger than the correlations between non-successive traits. Overall, the estimated moderate additive genetic variance in weaning weight, and the high genetic correlations between weaning weight and post-weaning market weight traits suggested that weaning weight could be used as a selection criterion for improving growth traits in the Mecheri sheep population studied.

## Supporting information

**S1 File. Dataset.**
(PDF)

## Acknowledgments

The authors are grateful for the facilities provided by the faculty of the Mecheri Sheep Research Station in Salem, India, and the higher authorities of the Veterinary College and Research Institute in Namakkal, India.

## Author Contributions

**Conceptualization:** Balakrishnan Balasundaram, Aranganoor Kannan Thiruvenkadan.

**Data curation:** Jaganadhan Muralidharan, Doraiswamy Cauveri.

**Formal analysis:** Balakrishnan Balasundaram, Aranganoor Kannan Thiruvenkadan.

**Funding acquisition:** Jaganadhan Muralidharan, Nagarajan Murali.

**Methodology:** Jaganadhan Muralidharan, Doraiswamy Cauveri.

**Resources:** Jaganadhan Muralidharan, Aranganoor Kannan Thiruvenkadan.

**Supervision:** Nagarajan Murali, Aranganoor Kannan Thiruvenkadan.

**Validation:** Moses Okpeku, Aranganoor Kannan Thiruvenkadan.

**Visualization:** Doraiswamy Cauveri.

**Writing – original draft:** Balakrishnan Balasundaram, Aranganoor Kannan Thiruvenkadan.

**Writing – review & editing:** Jaganadhan Muralidharan, Nagarajan Murali, Doraiswamy Cauveri, Angamuthu Raja, Moses Okpeku.

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
