## [Decision Letter · Decision Letter 0]

13 Nov 2023

PONE-D-23-22558Development of selection strategies for genetic improvement in production traits of Mecheri sheep based on a Bayesian multi trait evaluationPLOS ONE

Dear Dr. Thiruvenkadan,

Thank you for submitting your manuscript to PLOS ONE. After careful consideration, we feel that it has merit but does not fully meet PLOS ONE’s publication criteria as it currently stands. Therefore, we invite you to submit a revised version of the manuscript that addresses the points raised during the review process.

Dear Sir, 

Considering the reviewers' suggestions, 

I request that adjustments be made, to continue with the procedures. 

Best regards,

Julio Souza

We look forward to receiving your revised manuscript.

Kind regards,

Julio Cesar de Souza, Ph.D.

Academic Editor

PLOS ONE

Journal Requirements:

Whilst you may use any professional scientific editing service of your choice, PLOS has partnered with both American Journal Experts (AJE) and Editage to provide discounted services to PLOS authors. Both organizations have experience helping authors meet PLOS guidelines and can provide language editing, translation, manuscript formatting, and figure formatting to ensure your manuscript meets our submission guidelines. To take advantage of our partnership with AJE, visit the AJE website (http://aje.com/go/plos) for a 15% discount off AJE services. To take advantage of our partnership with Editage, visit the Editage website (www.editage.com) and enter referral code PLOSEDIT for a 15% discount off Editage services. If the PLOS editorial team finds any language issues in text that either AJE or Editage has edited, the service provider will re-edit the text for free.

"No" 

Additional Editor Comments:

Dear Sir,

Considering the reviewers' suggestions,

I request that adjustments be made, to continue with the procedures.

Best regards,

Julio Souza

Reviewers' comments:

Reviewer's Responses to Questions

**Comments to the Author**

1. Is the manuscript technically sound, and do the data support the conclusions?

Reviewer #1: Yes

Reviewer #2: Yes

2. Has the statistical analysis been performed appropriately and rigorously? 

Reviewer #1: Yes

Reviewer #2: Yes

3. Have the authors made all data underlying the findings in their manuscript fully available?

Reviewer #1: Yes

Reviewer #2: No

4. Is the manuscript presented in an intelligible fashion and written in standard English?

Reviewer #1: Yes

Reviewer #2: Yes

5. Review Comments to the Author

Reviewer #1: the paper title as well as the objective is very interesting. the methodology part was incorporated with wisely and correctly keep it up. statistical analysis part were better to do by use latest soft were.

Reviewer #2: Numbers of records used for the analysis should be clear in the abstract and methods sections. Although Table 1 has the numbers of records it is not clear if all the lamb records have an identified dam and sire. It would be clearer to the reader if "Dams (n=) and their progeny (n=) were analyzed. All lambs had identified sires (n=)".

Table 4 - model 2 is missing as is indication of best fit.

Discussion "birth weight" is birth weight a "growth factor"? For clarity suggest "birth weight is usually measured and used as an indication of growth potential"

Discussion "post-weaning growth traits" mid-paragraph. Suggest "Research on growth traits has established age-related increase in direct heritability..." Last sentence of the paragraph is long suggest breaking to two sentences. "The present study estimated moderate heritability values for post-weaning growth traits. This demonstrates there is adequate genetic variability for improved post-weaning growth through genetic selection."

6. PLOS authors have the option to publish the peer review history of their article (what does this mean?). If published, this will include your full peer review and any attached files.

Reviewer #1: No

Reviewer #2: **Yes: **Brenda Alexander

---

## [Author Response · Author response to Decision Letter 0]

27 Nov 2023

Response to Reviewers

We thank the academic editor and reviewers for the review and decision on regarding our submitted manuscript entitled, ‘Development of selection strategies for genetic improvement in production traits of Mecheri sheep based on a Bayesian multi trait evaluation’ (Ref. No. PONE-D-23-22558). 

Response to Academic Editor’s comments (Journal Requirements)

Review point 1. PLOS ONE's style requirements

Response: 

• Foot notes and legends under the tables were corrected as per the journal’s style. 

• Figure title and legends were corrected as per the journal’s style

• Equations were recreated with equation tools (MS word) as inline equations

• Mistakes in citing the order of the Reference within the text,[38] and [37] are corrected as the order of [37] and [38]

• Supporting information is included as per the style of the journal.

Review point 2. Language usage, spelling, and grammar

Response: 

The grammar, spelling, and usage of language throughout the entire document have been carefully reviewed and edited. The 'Revised Manuscript with Track Changes' file has corrections marked up. Dr. Moses Okpeku, who is from a native English-speaking nation, has reviewed the content.

Review point 3. Funding for this work

Response: We wish to mention that “the authors received no specific funding for this work”.

Review points 4 & 5. Data sharing 

Response: Data used in the analysis is anonymized and submitted as supporting information file along with the manuscript (S1_File). 

Review point 6. Captions for Supporting Information files

Response: Caption for the Supporting Information file is included at the end of the manuscript.

Review point 7. Reference list review 

Response: References in reference list and text have verified for their completeness and correctness.

Response to Reviewers’ comments

Reviewers comments Response 

Reviewer #1: the paper title as well as the objective is very interesting. The methodology part was incorporated with wisely and correctly keep it up. statistical analysis part were better to do by use latest software. The authors are thankful for the suggestion. We used the software program GIBBS2F90 in the data analysis and trust that this software is robust and relevant to the current study. 

Reviewer #2: Numbers of records used for the analysis should be clear in the abstract and methods sections – Please incorporate in abstract. Although Table 1 has the numbers of records it is not clear if all the lamb records have an identified dam and sire (do all lambs have identical dam and sire? If not, can we create a different table with number of lambs from the different sire (sire 2, sire 2 etc) and Dams (Dam1, Dam 2 etc)?. It would be clearer to the reader if "Dams (n=) and their progeny (n=) were analyzed. All lambs had identified sires (n=)" (Same as above). We thank the reviewer for critical assessment of the manuscript and their comments and the number of lambs with identified sires and dams, and number of dams and sires used to produce the lambs were included in the first paragraph of materials and methods section.

 The number of lamb records used were 2616, 2286, 1578, 1203 and 1019 for BW, BW3, BW6, BW9 and Bw12 respectively which is sired by 226, 208, 183, 167 and 160 rams for BW, BW3, BW6, BW9 and Bw12 and the number of dams belonging to the lambs records were 1044, 961, 814, 701 and 638 respectively.

Table 4 - model 2 is missing as is indication of best fit. The authors are grateful for the observation and the Model 2 has now been correctly entered in Table 4

Discussion "birth weight" is birth weight a "growth factor"? For clarity suggest "birth weight is usually measured and used as an indication of growth potential" The authors are grateful for this suggestion which has been effected accordingly, - Line number 239

Discussion "post-weaning growth traits" mid-paragraph. Suggest "Research on growth traits has established age-related increase in direct heritability..." Last sentence of the paragraph is long suggest breaking to two sentences. "The present study estimated moderate heritability values for post-weaning growth traits. This demonstrates there is adequate genetic variability for improved post-weaning growth through genetic selection."

 The authors appreciate the very useful suggestion, which has been effected accordingly, - Line numbers 296-303.

(A.K.THIRUVENKADAN)

---

## [Editor Report · Decision Letter 1]

1 Dec 2023

Development of selection strategies for genetic improvement in production traits of Mecheri sheep based on a Bayesian multi trait evaluation

PONE-D-23-22558R1

Dear Dr. Thiruvenkadan,

We’re pleased to inform you that your manuscript has been judged scientifically suitable for publication and will be formally accepted for publication once it meets all outstanding technical requirements.

Kind regards,

Julio Cesar de Souza, Ph.D.

Academic Editor

PLOS ONE
---

## [Editor Report · Acceptance letter]

5 Dec 2023

PONE-D-23-22558R1 

Development of selection strategies for genetic improvement in production traits of Mecheri sheep based on a Bayesian multi trait evaluation 

Dear Dr. Thiruvenkadan:

I'm pleased to inform you that your manuscript has been deemed suitable for publication in PLOS ONE. Congratulations! Your manuscript is now with our production department. 

Kind regards, 

on behalf of

Dr. Julio Cesar de Souza 

Academic Editor

PLOS ONE